# The Value of MRI and Radiomics for the Diagnostic Evaluation of Thyroid-Associated Ophthalmopathy

**DOI:** 10.3390/diagnostics15030388

**Published:** 2025-02-06

**Authors:** Weiyi Zhou, Yan Song, Jufeng Shi, Tuo Li

**Affiliations:** Department of Endocrinology, The Second Affiliated Hospital of Naval Medical University, Shanghai 200003, China; weiyizhou@smmu.edu.cn (W.Z.); songyanqf@163.com (Y.S.); shijufeng0513@126.com (J.S.)

**Keywords:** thyroid-associated ophthalmopathy, TAO, magnetic resonance imaging, MRI, radiomics, diagnosis

## Abstract

Thyroid-associated ophthalmopathy (TAO) is a vision-threatening autoimmune disease that involves the extraocular muscles (EOMs) and periorbital fat. Typical signs of TAO include eyelid recession, proptosis, diplopia, and decreased visual acuity. As a self-limited disease, there is major bipolarity in clinical outcomes in TAO population. The early diagnosis and prediction of these refractory and relapsed patients is essential. Unfortunately, commonly used tools such as CAS/NOSPECTS, are based on clinical symptoms and signs alone, have significant limitations. Some imaging techniques or examinations, such as magnetic resonance imaging (MRI), can be very effective in assisting TAO assessment, from exhaustive whiteboard notes to optimized patient outcomes. Being one of the most commonly used and accurate objective examinations for TAO assessment, MRI boosts no ionizing radiation, high soft tissue contrast, better reflection of tissue water content, and the ability to quantify multiple parameters. In addition, novel MR sequences are becoming increasingly more familiar in TAO and other areas of clinical and scientific research. Moreover, radiomics, a method involving the extraction of a large number of features from medical images through algorithms, is a more recent approach used in the analysis and characterization of TAO data. Thus, this review aims to summarize and compare the value of routine and novel functional MRI sequences and radiomics prediction models in the diagnosis and evaluation of TAO.

## 1. Introduction

Thyroid-associated ophthalmopathy (TAO) is a vision-threatening autoimmune disease affecting the extraocular muscles (EOMs) and periorbital fat. It is the most common extra-thyroidal manifestation of Graves’ disease (GD) and Hashimoto’s thyroiditis (HT) [1,2]. Graves’ disease is an inflammatory autoimmune state caused by thyrotropin (TSH) receptor autoantibody (TSHR-Ab). In GD, TSHR-Ab and the insulin-like growth factor-1 receptor (IGF-1R) autoantibody target their respective receptors on orbital fibroblasts and EOM groups, stimulating periorbital adipogenesis, persistent inflammation, and the final outcome of fibrosis. Orbital fibroblasts differentiate and proliferate into myofibroblasts and adipocytes, secreting hyaluronic acid and inflammatory cytokines. Persistent orbital inflammation leads to post-orbital adipogenesis and tissue edema, compressing the ocular vasculature and nerves and resulting in clinical symptoms such as proptosis, eyelid swelling, strabismus, and decreased visual acuity [3].

Orbital magnetic resonance imaging (MRI), with its high soft tissue contrast and multi-parameter quantification, has significant advantages over computed tomography (CT) or ultrasound in diagnosing and assessing TAO, especially the posterior inflammatory state of the orbits. MRI is free from ionizing radiation, has high intrinsic soft tissue contrast, reflects the tissue water content well, and can quantify multiple parameters [1] (see Figure 1).

Radiomics is a method of analyzing and interpreting diseases using imaging data. This method involves extracting a large amount of information from medical images that is not visible to the naked eye and uses advanced data analysis techniques to perform in-depth quantitative analyses of this information, enabling the non-invasive analysis of tissue heterogeneity. This approach enhances the understanding of the pathophysiological processes of disease and provides new perspectives on disease diagnosis, treatment, and prognosis in TAO [4].

## 2. Updates in MRI Sequences for TAO Diagnosis and Evaluation

In addition to routine sequences, novel functional MRI sequences have been applied in TAO assessment. Thus, we summarize regular and cut-edged MRI sequences and their imaging features, illustrating and comparing their application scenarios in TAO populations (Table 1).

### 2.1. Conventional Sequences

Conventional MRI sequences include T1-weighted imaging (T1WI) and T2-weighted imaging (T2WI). T2WI is used to evaluate edema and inflammatory infiltrates [1]. Short-tau inversion recovery (STIR) sequences selectively suppress the signal from fat while highlighting aqueous tissue. Inflammatory edematous EOMs appear as high signals in STIR sequences, making them highly discriminative in active TAO [1].

### 2.2. T1 Mapping

T1 mapping is a quantitative analytical technique that measures the T1 relaxation time of tissues, reflecting the nature of different tissues. Fibrosis often occurs in response to tissue damage caused by inflammatory stimuli. T1 mapping is a relatively new technique for assessing fibrosis, and an increased ΔT1 value positively correlates with the degree of histologic fibrosis, providing insights into EOM edema, fibrosis, and fatty infiltration [5].

### 2.3. T2 Mapping

T2 mapping is a quantitative analytical technique that measures the T2 values of tissues by quantifying the water content and collagenous tissue, reflecting changes in the extracellular fluid and collagen content [6]. T2 mapping not only measures the T2 relaxation time to quantitatively reflect the inflammatory state but also qualitatively assesses EOMs and adipose tissue [1].

### 2.4. Diffusion-Weighted Imaging (DWI)

DWI is a functional MRI technique that measures the degree of diffusion of water molecules. The apparent diffusion coefficient (ADC) quantifies the movement of water molecules in tissues, reflecting tissue-specific diffusion properties. The measurement of ADC is useful in quantitatively evaluating inflammation in the EOMs and plays an important role in the staging of TAO [5,6,7].

### 2.5. Dynamic Contrast-Enhanced (DCE) MRI

DCE-MRI shows the dynamics of body fluids after rapid contrast injection and dynamically observes the mode and degree of tissue enhancement in response to tissue perfusion, extracellular space volume, and vascular permeability. This technique provides vasculature-sensitive parameters and valuable information for assessing inflammatory diseases [5,8].

### 2.6. Diffusion Tensor Imaging (DTI)

DTI quantifies the diffusion of tissue water molecules in three-dimensional (3D) space by applying diffusion-sensitive gradients in multiple directions. DTI directly detects subtle microstructural changes in nerves by quantifying the microscopic movement of water molecules within nerve fibers. DTI is sensitive to potential structural changes in the nerve and has been widely used in the diagnosis of optic nerve diseases [9,10,11,12]. Fractional anisotropy (FA) is positively correlated with the density and integrity of the fiber, while mean diffusivity (MD) is negatively correlated with fiber bundle integrity.

### 2.7. Magnetization Transfer Image (MTI) and IDEAL-IQ

Magnetization transfer imaging (MTI) is an advanced MRI technique that provides additional information about water molecules bound to macromolecules (e.g., collagen) in the tissue of interest [13]. It indirectly reflects the concentration of macromolecules in the water physiological environment [14]. This technique can be applied to assess the degree of tissue fibrosis [15]. Iterative Decomposition to water and fat with Echo Asymmetry and Least-squares estimation-Intelligent Quantification (IDEAL-IQ) technology completely separates the water from fat tissue, overcoming the effects of inhomogeneous magnetic fields and achieving uniform fat suppression [16,17].

### 2.8. Radiomics

Radiomics extracts and analyzes numerous quantitative image details that cannot be detected by human experts. The non-invasive analysis of tissue heterogeneity is achieved by extracting quantitative features from radiographic images, including texture analysis and grayscale analysis [4,18].

## 3. Applications in TAO

### 3.1. Diagnosis of TAO

According to Bartley’s diagnostic criteria [19], it is important to evaluate thyroid function and thyroid-associated antibody levels, EOMs, proptosis, and the optic nerve. Clinical diagnosis is usually made by thorough history taking, physical examination, a combination of thyroid function and thyroid-related antibody tests, and imaging.

Of the three management strategies for hyperthyroidism (antithyroid medication, surgery, and radioactive iodine therapy), radioactive iodine therapy increases the risk of the development or exacerbation of Graves’ ophthalmopathy, although this risk can be mitigated by the use of a combination of steroid hormone therapy. Early recognition of concomitant TAO in patients with GD can avoid the risk of TAO development or progression that may be associated with radioactive iodine therapy [20].

Alterations in TAO on MRI imaging include ocular protrusion, increased retro-orbital fat, thickening of the EOMs, altered signaling of the EOMs, enlarged lacrimal glands (LGs), and swelling of the eyelids. Previous studies have focused on dilated EOMs and orbital adipose tissue, which are thought to be important contributors to ocular protrusion in TAO [21]. EOM involvement is widespread in patients with TAO, and both active and inactive patients may exhibit increased EOM volume [22]. MRI is widely used for the evaluation of EOM involvement, providing a quantitative and more objective visualization and assessment of EOM volume and thickness in patients with TAO. Normal EOMs show low signals on T1-weighted images and moderate signals on T2-weighted images on MRI. Kvetny et al. [23] found that the increased volume of EOMs is the main cause of ocular protrusion, emphasizing the importance of MRI-based EOM volumetric measurements. Nishida et al. [24] found that the most commonly involved muscle in patients with TAO is the inferior rectus and that the swelling of the inferior rectus and the medial rectus may be associated with limited eye movement. DWI can show EOM involvement earlier than conventional MRI. Studies have proven that the ADC values of the medial rectus and lateral rectus in individuals with TAO are higher than those in healthy populations, and the ADC value of the medial rectus can be used for the diagnosis of TAO [25,26,27,28], but lack harmonized cut-off values. The study by Liu et al. [9] found that combined histogram analysis and T2-mapping imaging could detect early pathology in EOM without conventional orbital MRI. This multiparametric approach may provide a breakthrough in exploring early damage to the EOMs in patients with TAO. Three-dimensional reconstruction is technically feasible in the diagnosis of patients with TAO, and Shen et al. [29] used Mimics software (developed by Materialise, Inc., headquarters in Leuven, Belgium) to successfully perform 3D reconstruction and measure the volume of the EOMs and retrobulbar connective tissue based on the MRI data. The findings showed that both the volume of orbital fat and the volume of EOMs were significantly increased in patients with TAO compared to those of healthy subjects.

The involvement of other periocular organs also plays a crucial role in TAO and is responsible for a wide range of clinical manifestations, with the LG being one of the most commonly involved organs. It was found that LG enlargement in patients with TAO occurs in parallel with the patient’s inflammatory activity and ocular discomfort [30,31,32,33]. Hu et al. [34] found that all quantitative measurements of the LG, except coronal length, were significantly greater in the TAO group than the healthy group based on routine serial quantitative MRI. A study by Wu et al. [5] showed that several parameters of the LG were significantly altered in TAO patients, with T2 and ΔT1 values being significantly higher in TAO patients than in GD patients; moreover, the Kep and Ve obtained using the DCE-MRI technique were also significantly smaller in the TAO group than the GD group. After multifactorial regression analysis, T2 and ΔT1 values were independent predictors for the diagnosis of TAO, and the diagnostic model constructed from T2 and Δ T1 values had an AUC of 0.94 (95%CI: 0.89–0.99, sensitivity of 81.82%, specificity of 100%) (see Figure 2). Razek et al. [35] found that the ADC values of the LG obtained from the DWI sequence were significantly higher in the TAO group than in the healthy control group. In the diagnosis of TAO, the critical value of ADC was 1.62 × 10^−3^ mm^2^/s with an AUC of 0.95 and an accuracy of 96% (see Figure 2). DTI is a favorable tool to show the microstructural changes of the LG. Chen et al.’s [36] study applied rs-EPI-based DTI examination to collect and compare the FA and ADC values of the LG. The LG-FA values of the TAO group were significantly lower than those of the healthy control group, while the LG-ADC values were much higher than those of the healthy control group (see Figure 2).

Dysthyroid optic neuropathy (DON) is the most serious complication of TAO, and the mechanism of DON is due to the enlargement of soft tissues (including the EOM and fat tissue), increased intraocular pressure, and immune inflammatory response, leading to orbital apical crowding. Previous imaging studies primarily used apical crowding signs (enlargement of EOMs, fat volume, and orbital bony geometry) and optic nerve stretching signs [37] to detect DON [17,38,39,40,41,42,43,44]. Various computational parameters of orbital bone and orbital soft tissue, including orbital volume, medial rectus muscle diameter, and EOM cross-sectional area and thickness, can be good predictors of DON [45]. However, studies have shown that the optic nerve is equally important for the diagnosis of DON; using high-resolution MRI to measure the diameter of the optic nerve in multiple planes revealed that the radial diameter of the optic nerve was useful in predicting the risk of DON [46]. In a study by Liu et al. [9], the DTI parameters of the optic nerve in each eye in the DON group and the non-DON group were computed and compared, including the mean, axial, and radial diffusivity (MD, AD, and RD, respectively) and anisotropy fraction (FA). The findings showed that MD alone had the best diagnostic efficacy, and the four parameters MD, AD, RD, and FA had the best combined diagnostic efficacy in differentiating between DON and non-DON. In a study by Wu et al. [17], changes in the optic nerve and cerebrospinal fluid in the optic nerve sheath were evaluated using the IDEAL-T2WI sequence in patients with DON, suggesting that an increase in the optic nerve and subarachnoid aqueous fraction 3 mm behind the globe was a reliable predictor of DON. Zhang et al. [45] demonstrated that the 3D reconstruction of the contours of the extraocular muscles, globe, optic nerve, and orbital fat revealed an increase in the volume of the extraocular muscles as a predictive factor for DON progression.

### 3.2. Staging of TAO Activity

The Rundle curve suggests that the natural course of TAO is a biphasic process that includes an active phase characterized by orbital inflammation and an inactive phase characterized by fibrosis and repair [1]. The accurate evaluation of the activity stage is important in the management of TAO. The Clinical Activity Score (CAS) is commonly used in clinical practice to assess the activity stage of TAO. The CAS scores include subjective symptoms and inflammatory signs, with a CAS of 3 or higher considered indicative of active TAO. However, CAS is a subjective scoring system [36] and observational variance between investigators can lead to greater heterogeneity in the assessment of the same patient [1,6].

MRI could serve as an adjudicative tool for activity staging, which allows multifaceted visualization and the quantitative measurement of EOMs, retro-orbital fat, LGs, and ocular prominence, making it an appropriate quantitative metric in addition to laboratory metrics. Conventional T2-based imaging with derived signal intensity (SI) or relaxation time is the most basic and widely recognized method and is considered a powerful tool for differentiating disease activity [47,48]. The signal intensity ratio (SIR) is the ratio of the SI of two tissues or EOMs, which are used in TAO assessments for standardization. It is positively correlated with the CAS and can be applied in TAO activity staging [49]. Edema of the orbital tissues and the infiltration of lymphocytes in the active phase are associated with elevated T2 values. DWI sequences are capable of quantifying the size and direction of aqueous spread, and their ADC values also show considerable potential in differentiating between active and inactive TAO [50]. In the active phase of EOM edema, the degree of diffusion is less restricted and ADC values are elevated, while in the inactive phase of EOM fibrosis, diffusion is significantly restricted and ADC values are reduced [51]. The STIR sequence selectively suppresses the signal from fat while highlighting the aqueous tissue, thus making it more discriminating in inflammatory edematous EOMs. Ge et al. [52] found that the SIR of EOMs on the STIR sequence was a good predictor of TAO activity and obtained a SIR threshold of 2.9 for the inferior rectus muscle. Orbital tissue fibrosis is also a pathologic change that can be of interest for activity staging, and T1 relaxation time can assess EOM fibrosis as well as fat infiltration [53,54,55]. Hu et al. [15] found that the magnetization transfer ratio (MTR) obtained through MTI sequencing could distinguish between active and inactive patients, and the negative correlation between MTR and CAS may further indicate that the degree of fibrosis increases with decreasing disease activity, demonstrating good discriminatory properties and reproducibility. The degree of LG involvement can, likewise, be suggestive of activity stage, and a study by Gagliardo et al. [56] found that the parameter herniation of LG (LGH), obtained using simple quantitative measurements suggestive of the herniation of LG was significantly higher in active than inactive patients, suggesting that TAO activity staging can be performed using LG herniation. A study by Chen et al. suggested that the LG-FA and ADC obtained from Rs-EPI-based DTI examination could help in the staging of TAO activity, with LG-FA values being significantly lower in the active stage than in the inactive stage and LG-ADC values being significantly higher than in the inactive stage. In differentiating TAO activity, FA values showed significantly higher AUC than ADC values (0.832 vs. 0.570, *p* = 0.009) [36] (see Figure 2). A study by Razek et al. [35] found that the DWI sequence-based ADC values of LG were significantly higher in the active patients than in the inactive patients and that the ADC values were positively correlated with CAS (r = 0.78, *p* = 0.001) with an AUC of 0.80 (accuracy of 82%, sensitivity of 65%, specificity of 100%, positive predictive value (PPV) of 100%, and negative predictive value (NPV) of 73%) (see Figure 2).

### 3.3. Evaluation and Prediction of Efficacy

For active and moderately severe TAO patients, the recommended first-line treatment is high-dose intravenous glucocorticoid (IVGC) therapy [16]. However, 20–40% of patients fail to respond to this first-line treatment, and there is an equal risk of adverse effects such as secondary hypertension, hyperglycemia, hepatic impairment, and osteoporosis, which can seriously affect their quality of life [57,58,59]. Thus, identifying TAO patients who truly benefit from IVGC before therapeutic administration is necessary to avoid ineffective treatment. Previous studies have shown that the predictive accuracy is insufficient when based on empirical clinical judgment alone, such as pre-treatment response and smoking history [60,61]. The combination of clinical metrics with MRI-based quantitative measurements of periorbital tissue may enhance the efficacy of predictive models.

The treatment of TAO is aimed at suppressing orbital inflammation and reducing the subsequent tissue remodeling of periocular soft tissues such as EOMs and orbital fat [62]. MRI and radiomic approaches can reveal inflammatory changes in the orbital fat and EOMs in patients with TAO [63] and can be used to objectively and quantitatively assess the level of orbital inflammation, changes in EOMs and periorbital soft tissues such as periorbital fat, and the efficacy of the treatment.

MRI images can be utilized to observe changes in proptosis, EOM volume, and retro-orbital fat volume to assess the effectiveness of treatment [64]. EOM volume, fat volume, and SIR are considered to be the basic quantitative indexes for assessing the efficacy of treatment. A study by Chen et al. [65] found significant differences in SIR and EOM at SI_max_ before and after RTX treatment. Jain [63] et al. found that EOM volume and orbital fat volume were reduced in TAO patients before and after teprotumumab treatment; orbital fat was less responsive to teprotumumab treatment, and the SIR values were significantly lower in TAO patients after treatment. Other functional sequences or particular parameters have also been used to evaluate validity. Yu et al. [25] found that the ADC values of EOMs on DWI functional sequences were significantly lower in TAO patients before and after receiving radiation therapy, suggesting that ADC values can be a quantitative indicator for assessing response to treatment. A study by Duan et al. [66] observed that, in TAO patients with upper lid retraction, the SI of the levator palpebrae superioris (LPS) muscle was significantly lower in the tretinoin injection treatment-effective group than in the treatment-ineffective group, and also found that the pre-treatment SIR of the LPS muscle on enhanced T1WI sequences had a better predictive value of the treatment effect. It has also been suggested that TAO eyelid abnormalities cannot be explained qualitatively simply by LPS muscle thickening. The degree and location of LPS muscle thickening and impaired LPS muscle function may be the key determinants of the clinical presentation of the eyelids. Higashiyama et al. [49,67] found that the SIR and volume of all EOMs decreased significantly after methylprednisolone treatment on STIR sequences, with obvious positive correlations between SIR and volume. Thus, the SIR reflects the degree of inflammation of muscles, and a higher SIR suggests the risk of worsening in TAO. Therefore, the SI of EOM is suggested to be a sensitive index for evaluating the efficacy of TAO treatment. T2 relaxation time is also a possible quantitative indicator of treatment efficacy. Tachibana’s study [68] found that the mean T2 relaxation time and the maximum T2 relaxation time were significantly reduced in TAO patients after immunotherapy.

Zhu et al. [54] measured and compared raw T1 values and post-enhancement T1 values of the EOM and found that the native T1 of the superior rectus muscle was an independent predictor of GC treatment responsiveness. Hu’s study [69] retrospectively collected and compared pre-treatment MRI-based EOM, orbital fat tissue (OFT), and LG parameters, as well as clinical factors in patients from treatment-responsive and non-responsive groups, and found that the conventional sequence EOM-SIR min and the LGH/OFT ratio can be used as promising markers for predicting the response to IVGC treatment in patients with active and moderate-to-severe TAO. In a study by Zhai et al. [16], T2 IDEAL was applied to analyze and compare the baseline clinical characteristics and imaging parameters of two different groups of patients with different efficacy. Here, EOM-T2 relaxation time (T2RT)_mean_ and EOM-Water Fraction (WF)_max_ were found to be independent predictors of TAO patients responding to GC therapy (see Figure 2). Amy et al. [63] evaluated teprotumumab efficacy in TAO, finding that the total EOM volume was significantly reduced in all patients after treatment (mean reduction 33%, *p* < 0.02) and the EOM-SIR was reduced (*p* < 0.01) in teprotumumab trials.

Based on the radiomics of orbital MRI images, the histogram features [70] of the EOMs and the texture features [18] of the EOMs and retro-orbital fat before treatment can predict whether IVGC treatment is effective or not, which can be a useful clinical application to guide the treatment plan for patients with TAO and to advance the precision of medical treatment. In the study by Hu et al., the radiomics of patients receiving systemic intravenous glucocorticoid was investigated using MRI before and after the treatment based on coronal T2-weighted fat suppression imaging. Hu et al. used radiomics to extract features from MRI images of patients before and after receiving systemic IVGC, establishing a predictive model of the treatment effect. The efficacy of this model was compared with the model established using semi-quantitative measurements, and the EOM imaging model based on T2WI showed better diagnostic efficacy than the semi-quantitative model [4] (see Figure 2).

## 4. Conclusions and Prospects

The application of imaging and radiomics in TAO demonstrates their potential in assisting diagnosis, disease assessment, evaluation of treatment, and prognosis (see Figure 1). However, the application of radiomics in TAO is still in its infancy, and further research and practice are needed to improve and expand its application. In the future, with the continuous development of imaging technology and the innovation of data analysis methods, we expect radiomics to play a greater role in the diagnosis, treatment, and prognosis of TAO and other autoimmune diseases.

## Figures and Tables

**Figure 1 diagnostics-15-00388-f001:**
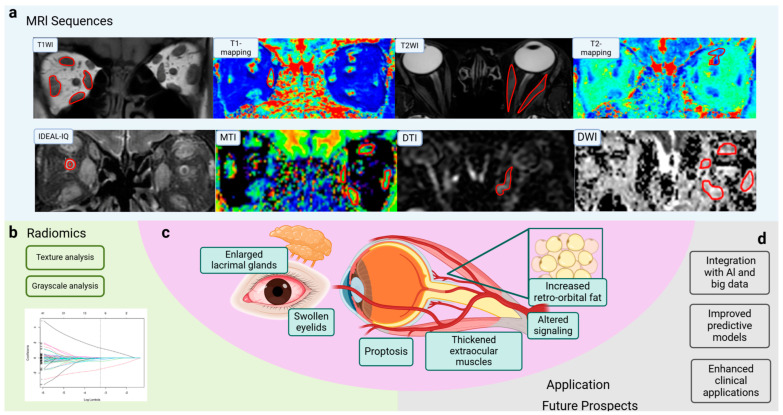
Application of MRI sequences and radiomics in TAO. (**a**) Novel MRI sequences such as T1 mapping, T2 mapping, IDEAL-IQ, and DWI are used for TAO evaluation and obtain different dimensions of information. (**b**) Characteristics of radiomics can contribute to TAO assessment. (**c**) Symptoms of TAO, such as enlarged lacrimal glands, swollen eyelids, and proptosis, are associated with thickened extraocular muscles, altered signaling, and increased retro-orbital fat. (**d**) MR imaging has been widely applied in disease diagnosis, staging and evaluation, and prognosis prediction. Integration with AI and big data analysis improves predictive models and enhances clinical applications. MRI, magnetic resonance imaging; T1WI, T1-weighted imaging; T2WI, T2-weighted imaging; IDEAL-IQ, Iterative Decomposition to water and fat with Echo Asymmetry and Least-squares estimation-Intelligent Quantification; DWI, diffusion-weighted imaging; DTI, diffusion tensor imaging; MTI, magnetization transfer imaging; SIR, signal intensity ratio; EOM, extraocular muscle; AI, artificial intelligence; TAO, thyroid-associated ophthalmopathy. (Created in BioRender. Zhou, W. (2025) https://BioRender.com/c78e623).

**Figure 2 diagnostics-15-00388-f002:**
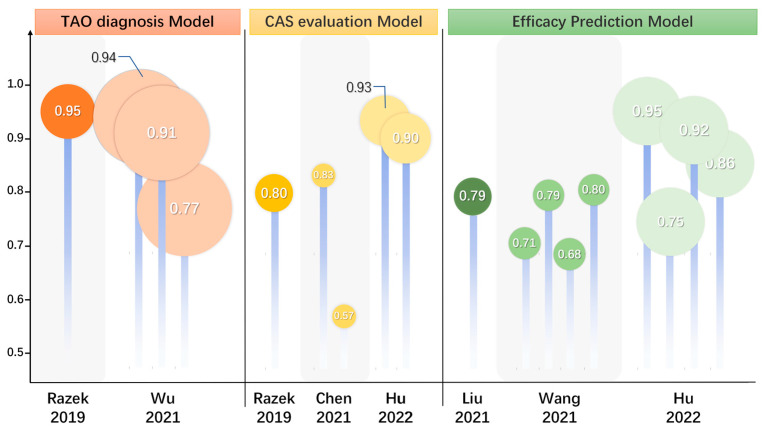
Updates and comparison of efficacy on diagnosis, evaluation, and prognosis prediction in TAO. The summary and comparison of statistical efficacy of MRI predictive models in seven studies. These models were used for TAO diagnosis, CAS activity staging, and treatment efficacy prediction. TAO, thyroid-associated ophthalmopathy; MRI, magnetic resonance imaging; CAS, clinical activity score. [4,7,17,18,35,36].

**Table 1 diagnostics-15-00388-t001:** Summary of novel MRI sequences and applications in TAO.

Sequences	Applications in TAO	Description	Advantages
STIR	Assessment of activity stage	Selective suppression of fat signals, highlights water tissue	Highly discriminatory for inflammatory edema
T1 mapping	Evaluation of the degree of fibrosis	Detects small changes in water molecules, proteoglycans, collagen content, etc. in tissues	Excellent scanning consistency and reproducibility
T2 mapping	Assessment of activity stage	Quantification of tissue moisture and collagen fiber content and composition	Excellent scanning consistency and reproducibility
DWI	Assessment of activity stage	Quantifying the movement of water molecules in tissues	Reduction of magnetic susceptibility artefacts and geometrical variations, improving the signal-to-noise ratio of orbital images
DCE-MRI	Assessment of activity stage, prediction of treatment response to IVGC	Provides quantitative information about the microcirculatory perfusion and permeability of various tissues	Shows further physiological alterations within EOMs
DTI	Assessment of activity and severity stage	A quantitative method for observing the anisotropy of water molecule diffusion in tissues	Elimination of bias in estimating the microstructure of tissues and the ability to distinguish changes in the tissue itself
MTI	Evaluation of the degree of fibrosis	Provides additional information about water molecules bound to macromolecules	Detects fibrotic changes of orbital tissues
IDEAL-IQ	Assessment of activity and severity stage, prediction of treatment response to IVGC	Enables the quantitative measurement of water and lipid content in tissues, respectively	Evaluates both inflammation and fibrosis
Radiomics	Assessment of activity and severity stage, prediction of treatment response to IVGC	Extracts high-throughput quantitative features from medical images that reflect tissue and lesion characteristics such as heterogeneity and shape	Large number of quantitative features, can be fused multimodally

TAO, thyroid-associated ophthalmopathy; STIR, short-tau inversion recovery; DWI, diffusion-weighted imaging; DCE-MRI, dynamic contrast-enhanced (DCE) MRI; EOM, extraocular muscle; DTI, diffusion tensor imaging; MTI, magnetization transfer imaging; IDEAL-IQ, Iterative Decomposition to water and fat with Echo Asymmetry and Least-squares estimation-Intelligent Quantification.

## Data Availability

Not applicable.

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
