# Peer review of "The Value of MRI and Radiomics for the Diagnostic Evaluation of Thyroid-Associated Ophthalmopathy"

_diagnostics, 2025, doi:10.3390/diagnostics15030388_

Round 1

Reviewer 1 Report

Comments and Suggestions for Authors

The review article aims to describe the value of MRI and radiomics in the diagnosis  and evaluation of TAO. However they should provide more images examples to show their advantages.  

1.          The authors address the advantage of DWI can show extraocular muscle involvement earlier than conventional MRI. In addition, ADC values of the rectus medialis and rectus externus in TAO pop ulation are higher than those in the healthy ones, and the ADC value of the rectus medialis can be used for the diagnosis of TAO. Could authors provide these MRI images examples?

2.          Please also provide image of MRI and/or examples of Radiomics to reveal early diaggosis of dysfunctional optic neuropathy (DON)

3.          Please also provide solid image of MRI/radiomics to show staging of TAO activity or prediction of outcome.

Author Response

Comments 1: The authors address the advantage of DWI can show extraocular muscle involvement earlier than conventional MRI. In addition, ADC values of the rectus medialis and rectus externus in TAO population are higher than those in the healthy ones, and the ADC value of the rectus medialis can be used for the diagnosis of TAO. Could authors provide these MRI images examples?”

Response 1: We appreciate for the 1st reviewer’s comments, we agree with this comment. Therefore, we have added MRI images of the extraocular muscle ADC values for the evaluation of TAO in Figure 1a (at P2-L55) and supplemented ADC values and cut-off values in TAO patients in several studies at P2-L149 to L152. However, the authorized cut-off values in ADC values are still needed for more studies.

Comments 2: “Please also provide image of MRI and/or examples of Radiomics to reveal early diagnosis of dysfunctional optic neuropathy (DON)”.
Response 2: Thanks for the 1st reviewer suggestions. We agree with this comment. However, we haven’t found any study related to Radiomics focused on DON. Thus, we just added novel MRI sequences such as DTI and IDEAL-IQ for DON assessment associated images to figure 1a (at P2-L55) and described the representative studies at P5-L193 to L201.

Comments 3: “Please also provide solid image of MRI/radiomics to show staging of TAO activity or prediction of outcome.”

Response 3: Thanking for the 1st reviewer’ s advice on supplementary of images of MRI applications for TAO stage. We rewrote and polished the 3.2.PART --Staging of TAO activity at P6-L219 to L237 for diagnosis and prediction of TAO via novel sequence and radiomics model. Besides, we also added the MTI sequence to figure 1a (at P2-L55) and illustrated at P6-L233 to L237.

Reviewer 2 Report

Comments and Suggestions for Authors

Unfortunately, the text contains numerous spelling and grammatical errors. It should undergo thorough grammar and language editing by a native speaker. Other than these issues, I did not notice any major flaws. Presenting the mentioned imaging sequences in a table summarizing their advantages, disadvantages, and intended use would make the manuscript more comprehensible. The figures currently provide very general information; however, examples of findings from the different sequences discussed in the text are missing. Including a few example figures would enhance the manuscript.

You can find my detailed corrections and comments in the attached PDF file. I would be happy to review the revised version based on these suggestions.

Comments on the Quality of English Language

Unfortunately, the text contains numerous spelling and grammatical errors. It should undergo thorough grammar and language editing by a native speaker.

Author Response

Comment 1: “The text contains numerous spelling and grammatical errors. It should undergo thorough grammar and language editing by a native speaker.”
Response 1: Thank you for pointing this out. We rewrote the manuscript for spelling corrects and asked for MDPI English editing for best understood.

Comment 2: Presenting the mentioned imaging sequences in a table summarizing their advantages, disadvantages, and intended use would make the manuscript more comprehensible.”
Response 2: Thanks for the 2ndreviewer’s suggestion. We assembled and compared the advantages and potential applications of each sequence in paragraph at P2-L66 to L69 (in highlight) and detailed in Table 1 at P12-L369.

Comment 3:  The figures currently provide very general information; however, examples of findings from the different sequences discussed in the text are missing. Including a few example figures would enhance the manuscript.”
Response 3: Thanks for the 2nd reviewer’s suggestion. We have added MRI images of the extraocular muscle ADC values for the evaluation of TAO in Figure 1a (at P2-L55) and supplemented at P2-L149 to L152, added the novel MRI sequences such as DTI and IDEAL-IQ for DON assessment associated images in Figure 1a (at P2-L55) and description at P5-L193 to L201, the MTI images for TAO assessment in Figure 1a (at P2-L55) and description at P6-L233 to L237.

Comment 4: Please rephrase sentences to better align with the distinct purpose of the abstract.”
Response 4: Thanking for the 2nd reviewer’s request. We rephrase the abstract that were identical to those in the main text at P1-L9 to L26 (in highlighted).

Comment 5: “You’d better provide the full names of the abbreviations in the figure 1, name the groups separately and to modify the legend accordingly”.
Response 5: Thanking for the 2nd reviewer’s suggestion. We revised figure 1, added MRI images as examples, revised figure notes according to instruction, and added the full names of the abbreviations in the figure 1 footnotes (at P2-L56 to L64).

Comment 6: “Provide more information about DWI technique and its use in MRI of TAO patients.”
Response 6: Thanks for the 2nd reviewer’s suggestion. We added description of the use of DWI sequence and ADC value in TAO population which is highlighted at P3-L91 to L94  and added related references [5-7].

Comment 7: The 2nd reviewer asked us to clarify the part that led to the confusion.
Response 7: We apologize for the confusion to the 2nd reviewer. We revised the sentences, deleted difficult parts, and clarified in details.  They are:
(1) P4-L165: “quantitative measurements...”. After revision, “all quantitative measurements of the LG...”.
(2) P6-L219 to L221: “The SIR of the extraocular and ipsilateral temporal muscles correlates positively with the CAS score and can be applied to TAO activity staging...”. After revision, “The signal intensity ratio (SIR) is the ratio of the SI of two tissues or EOMs, which are used in TAO assessments for standardization. It is positively correlated with the CAS and can be applied in TAO activity staging...”.
(3) P6-L233 to L234: “MTRs of MTI...”. After revision, “the magnetization transfer ratio(MTR) obtained through MTI sequencing...”.
(4) P6-L247: “the DWI sequence-based ADC values...”. After revision, “the DWI sequence-based ADC values of LG...”.

Comment 8: The 2nd reviewer asked us to modify the presentation of the sentence.
Response 8: Thanks for the 2nd receiver's suggestion. We have revised the sentences in according to the guidance. The revisions are showed in highlighted words:
(1) P5-L201 to L204: “Founded by Zhang et al, the 3D reconstruction of the contours of the extraocular muscles, the globe, the optic nerve, and the orbital fat found that an increase in the volume of the extraocular muscles was a predictive factor for DON progression...”. After revision, “Zhang et al. demonstrated that 3D reconstruction of the contours of the extraocular muscles, globe, optic nerve, and orbital fat revealed an increase in the volume of the extraocular muscles as a predictive factor for DON progression....”.
(2) Figure 2 footnote: “Summary and compare the statistical efficacy of MRI predictive models in 7 researches. This models were used to TAO diagnosis, CAS activity staging and treatment efficacy prediction...”. After revision, “The summary and comparison of statistical efficacy of MRI predictive models in seven studies. These models were used for TAO diagnosis, CAS activity staging, and treatment efficacy prediction...”.

Reviewer 3 Report

Comments and Suggestions for Authors

1. There are several errors in the use of abbreviations. For example, the authors used the abbreviation "MRI" which is introduced in line 36 before it is defined in line 38. Similarly, the definition of "EOM" should be placed in line 25, not in line 128.

2. The term "thyroid dysfunctional optic neuropathy" is uncommon and should be changed to "dysthyroid optic neuropathy."

3. Suggest to review and revise the abstract. It is informative but somehow lacks content about the study's main contributions and outcomes. It reads more like an introduction than a summary of the findings.

4. In lines 32–33, persistent inflammation also leads to fibrosis.

5. In lines 172–174, I recommend that the authors include other mechanisms for the development of dysthyroid optic neuropathy.

6. In line 217, I think that fatty infiltration is associated with a long T2 value.

7. In line 310, I recommend that the authors add information on the effectiveness of teprotumumab as confirmed by MRI. Please read the following paper: doi: 10.1136/bjophthalmol-2020-317806.

8. How will radiomics and advanced MRI techniques impact patient outcomes, particularly in resource-limited settings?

Author Response

Comment 1: “There are several errors in the use of abbreviations. For example, the authors used the abbreviation "MRI" which is introduced in line 36 before it is defined in line 38. Similarly, the definition of "EOM" should be placed in line 25, not in line 128. ”
Response 1: Thanks for pointing these out. We apologized for our misuse of abbreviations, thank for the 3rd reviewer’s guidance. We have fixed all the errors in the use of abbreviations with Red colored words:
(1) EOM: P1-L31
(2) MRI: P1-L42
(3) CT: P1-L44
(4) IDEAL-IQ: P3-L114 to L115
(5) LG: P4-L136
(6) LGH: P6-L239
(7) OFT: P7-L299
(8) SIR: P6-L219

Comment 2: The 3rd reviewer asked us to correct the term of DON: The term "thyroid dysfunctional optic neuropathy" is uncommon and should be changed to "dysthyroid optic neuropathy.
Response 2: Thanks for the 3rd reviewer’s comments. We correct the term of DON to dysfunctional optic neuropathy at P5-L182 and abbreviations list.

Comment 3: The 3rd reviewer suggests us to review and revise the abstract: “Suggest to review and revise the abstract. It is informative but somehow lacks content about the study's main contributions and outcomes. It reads more like an introduction than a summary of the findings.”
Response 3: Thanks for the 3rd receiver's guidance. We rephrase sentences that used in the main text, and add content about the study's main contributions and outcomes at P1-L9 to L26 (in highlighted).

Comment 4: The 3rd reviewer suggests: In lines 32–33, “persistent inflammation also leads to fibrosis”.
Response 4: Thanks for the 3rd receiver's guidance. According to 3rd reviewer’s comments, we added “fibrosis” as the final outcome of TAO patients at P1-L37, the revision is:
In GD, TSHR-Ab and the insulin-like growth factor-1 receptor (IGF-1R) autoantibody target their respective receptors on orbital fibroblasts and EOM groups, stimulating periorbital adipogenesis, persistent inflammation, and the final outcome of fibrosis.

Comment 5: The 3rd reviewer recommend that “the authors include other mechanisms for the development of dysthyroid optic neuropathy.”
Response 5: Thanks for the 3rd reviewer’s comments. In accordance with the 3rd reviewer’s suggestion, we supplied some new mechanisms lead to DON in GD patients at P5-L184. The revision is:
Dysfunctional optic neuropathy (DON) is the most serious complication of TAO, and the mechanism of DON is due to the enlargement of soft tissues (including the EOM and fat tissue), increased intraocular pressure, and immune inflammatory response, leading to orbital apical crowding.

Comment 6: The 3rdreviewer suggestion us that “the performance of long T2-relaxiation value of fatty infiltration”
Response 6: We appreciate for the 3rd reviewer’s comments. Meanwhile in recent studies, it is generally acknowledged that T2RT of fatty infiltration is reduced in most situations, and we searched for references with the same performance (DOI: 10.1002/nbm.4111). While, there is still other possibility of T2RT performances in fatty infiltration, thus we deleted the controversial sentence and rewrote it at P6-L221 to L223:
Edema of the orbital tissues and the infiltration of lymphocytes in the active phase are associated with elevated T2 values.

Comment 7: The 3rd reviewer recommended us to add information on the effectiveness of teprotumumab as confirmed by MRI: “Please read the following paper: doi: 10.1136/bjophthalmol-2020-317806”.
Response 7: Thanks for the 3rd reviewer’s suggestion. The performance of MRI to evaluate the efficacy of teprotumumab phase trial was added the ref [70] at P7-L307.

Comment 8: The 3rd reviewer asked “How will Radiomics and advanced MRI techniques impact patient outcomes, particularly in resource-limited settings?”
Reply: Thanks for the 3rd reviewer’s suggestion. The cut-edged method for diagnosis and evaluation of the TAO patients is with great clinical importance but lacks. Besides, the MRI related multi-meters models in some studies even show their promising prediction of TAO patients’ prognosis and medication choice. Thus, we summarized the recent novel sequences, indices and radiomics models for literature review. And we have added an explanation for this issue.

Round 2

Reviewer 1 Report

Comments and Suggestions for Authors

no more comment

Author Response

No more comment from the 1st Reivewer.

Reviewer 2 Report

Comments and Suggestions for Authors

I'd like to thank the authors for their meticulous effort for the reviewing process. I believe that the revised version of the manuscript is suitable for publication.

Author Response

No more comments from the 2nd Reviewer.
We are very grateful to the reviewer‘s professional guidance.

Reviewer 3 Report

Comments and Suggestions for Authors

The expanded form of the abbreviation "DON" is usually "dysthyroid optic neuropathy", not dysfunctional optic neuropathy.

Author Response

Comment 1: The 3rd Reviewer request us : "The expanded form of the abbreviation "DON" is usually "dysthyroid optic neuropathy", not dysfunctional optic neuropathy".
Response: Thanks a lot for poninting this out. As the revierwer's comment, we corrected the explanded form of DON to "dysthyroid optic neuropathy" in hgihlighted at P5-L187 and abbrevation list at P8-L342.